# Tailoring Tofacitinib Oral Therapy in Rheumatoid Arthritis: The TuTOR App

**DOI:** 10.3390/ijerph19095379

**Published:** 2022-04-28

**Authors:** Massimo Radin, Marta Arbrile, Irene Cecchi, Pierluigi Di Nunzio, Nicola Buccarano, Federico Di Gregorio, Valeria Milone, Sara Osella, Paola Crosasso, Marika Denise Favuzzi, Alice Barinotti, Simone Baldovino, Elisa Menegatti, Daniela Rossi, Savino Sciascia, Dario Roccatello

**Affiliations:** 1Center of Excellence for Nephrologic, Rheumatologic and Rare Diseases, CMID-Nephrology and Dialysis, Department of Clinical and Biological Sciences, Ospedale San Giovanni Bosco, 10154 Turin, Italy; massimo.radin@unito.it (M.R.); marta.arbrile@unito.it (M.A.); irene.cecchi@unito.it (I.C.); simone.baldovino@unito.it (S.B.); daniela.rossi@unito.it (D.R.); dario.roccatello@unito.it (D.R.); 2Department of Clinical and Biological Sciences, School of Specialization of Clinical Pathology, 20122 Turin, Italy; marikadenise.favuzzi@unito.it (M.D.F.); alice.barinotti@unito.it (A.B.); elisa.menegatti@unito.it (E.M.); 3DNDG s.r.l., 10100 Turin, Italy; gg@dndg.it (P.D.N.); nico@dndg.it (N.B.); fog@dndg.it (F.D.G.); 4Pharmacy Department, S. Giovanni Bosco Hospital, 10154 Turin, Italy; valeria.milone@aslcittaditorino.it (V.M.); sara.osella@aslcittaditorino.it (S.O.); paola.crosasso@aslcittaditorino.it (P.C.)

**Keywords:** rheumatoid arthritis, adherence, mobile app, tofacitinib, digital health

## Abstract

To support the management of rheumatoid arthritis (RA) patients treated with tofacitinib, we designed the TuTOR (tailoring tofacitinib oral therapy in rheumatoid arthritis) mobile app. The impact of the app on medical adherence was evaluated using a crossover design alternating a paper-diary and the TuTOR App. Twenty patients with RA (mean age at inclusion, 59 ± 13 years) were included in the study. A statistically significant decrease in DAS28 was observed since the first month of therapy (mean DAS28 at baseline, 3.9 ± 1 vs. 1° month 3.1 ± 1, *p* = 0.0016). Similarly, the numerical rating scale (NRS) of perceived activity of disease and subjective fatigue progressively decreased. No differences were reported in DAS28 or NRS between the TuTOR app and the paper-diary groups. A significant decrease was observed in HAQ during the follow-up (baseline 1.38 ± 1.11 vs. six months 0.83 ± 0.9; *p* = 0.01). When filling out the self-reporting questionnaires, most of the patients (82%) preferred the TuTOR App helping them to remember to take the pills. Furthermore, 82% of patients used the app regularly (vs. 53% for the paper diary). Three patients suspended tofacitinib due to gastrointestinal intolerance. Both digital and paper devices can help maximize adherence to therapy; however, the TuTOR app was preferred by the patients for its simplicity and immediacy.

## 1. Introduction

Rheumatoid arthritis (RA) is a chronic autoimmune systemic inflammatory disease affecting between 0.5% and 1% of the adult population worldwide [1,2,3,4]. Over the years, evolutions in the understanding of the signaling pathways downstream of many proinflammatory cytokines have led to the development of biological disease-modifying anti-rheumatic drugs (bDMARDs) that profoundly changed how clinicians approach RA patients [5,6]. Research in this field is ever-evolving, and new molecules are constantly being developed to improve RA management [7]. An example can be found in tofacitinib, a Janus kinase inhibitor that has recently been approved as a treatment option in patients with RA [8,9]. One of the most interesting peculiarities of tofacitinib, when compared to most biological drugs for RA, is its oral administration. Although from the patient’s perspective, oral administration is considered an advantage, adherence to self-management and medication might raise some concerns among treating physicians. Medication non-adherence has a significant impact on the health and well-being of individuals with chronic diseases. With respect to important risk factors of RA, such as cardiovascular risk factors, it is known that up to 50% of patients stop taking medication for these conditions during the first year of prescription [10]. A wide variety of factors can be recognized as reasons for non-adherence to medical prescriptions, including psychological and sociocultural barriers that should be addressed with a patient-centered approach. In recent decades, the enormous development of digital technologies has made available a huge number of new tools, e.g., smartphone applications and tablets, that are useful for improving patient management. Among them, the employment of telemedicine has become particularly relevant during the COVID-19 pandemic for the follow-up of chronic patients requiring constant monitoring [11,12]. To support the management of RA patients treated with tofacitinib, we designed a mobile application: the TuTOR (tailoring tofacitinib oral therapy in rheumatoid arthritis) mobile app. As a first part of the study, the TuTOR app and the methodology of the study were designed, as previously described [13]. In this second study, we aimed to evaluate the impact of the TuTOR app on medical adherence by performing a prospective controlled, single-center study. Our primary objective was to validate the use of the TuTOR app in a controlled population of RA patients and, secondly, to determine whether the use of a mobile application could ameliorate adherence to therapy. 

## 2. Materials and Methods

### 2.1. Patients Characteristics

#### 2.1.1. Inclusion Criteria

Patients included in the study included individuals over the age of 18 who meet the 2010 ACR and EULAR classification criteria for RA [14], with active disease defined as having 4 or more tender or painful joints on motion and 4 or more swollen joints at baseline despite treatment with methotrexate (15–25 mg per week), high-sensitivity C-reactive protein (CPR) of ≥3 mg/L and class I–III functional capacity as classified by the ACR 1991 revised criteria for global functioning status in RA [15]. Patients were required to discontinue all conventional synthetic (cs) DMARDs other than methotrexate for at least 4 weeks before baseline and continue to receive, if needed, stable non-steroidal anti-inflammatory drugs, analgesics and/or oral corticosteroids throughout the duration of the study. Patients who had responded inadequately or had an adverse event with a bDMARD were included after having discontinued the bDMARD for a minimum period of time depending on the specific agent (i.e., rituximab, 52 weeks; abatacept and tocilizumab, 12 weeks; golimumab, 10 weeks; anakinra and etanercept, 4 weeks).

#### 2.1.2. Exclusion Criteria

Contraindications for any study treatment included history of infections requiring treatment within 2 weeks or any admission to hospital within the 6 months before the beginning of the study; exclusionary morbidities, HIV, hepatitis B or C; inadequately treated or undocumented treatment for tuberculosis; one episode of disseminated herpes zoster or herpes simplex; any clinically significant laboratory abnormalities; and ongoing pregnancy and lactation. Patients who had previously received Tofacitinib or live attenuated vaccines other than the herpes zoster vaccine within 6 weeks before study initiation were also excluded.

### 2.2. Study Design

This study included a total of 20 RA patients who began treatment with tofacitinib jointly with the use of the TuTOR mobile app or with the use of a paper diary (PD). The design of the study was prospective, controlled and single-center. All patients were evaluated at baseline to assess the affinity for technology and its influence on the outcomes of the study through the medium of certified questionnaires based on digital health literacy. Subsequently, they received a personal learning-by-doing tutorial session in order to familiarize them with the functions of the application (i.e., confirming medication intake and recording follow-up visits and clinical appointments). Appendix A shows the baseline educational level, affinity with technology and personal information of the patients enrolled in the study. The study was conducted using a crossover design with 3 sequences: the initial phase (training and baseline assessment), the interventional phase (3 months of therapy using the App system), and the comparative phase (3 months of treatment using a PD). Users experienced the interventional and comparative phases alternately. Figure 1 shows the timeline of the research protocol of the study.

All participants were followed-up for 6 months, with a clinical ambulatory visit each month and personal adherence questionnaires administered using a paper format at baseline and every 3 months, as previously described [13]. Tablets were supplied in blisters of 56 film-coated tablets. At each visit, patients were required to bring their medication blisters; the number of remaining tablets was recorded by the clinician and compared with the self-reported adherence. During the monthly ambulatory visits, the subjective levels of fatigue, disease activity and pain were evaluated using a numerical rating scale (NRS); disease activity score 28 was also regularly assessed [16,17]. Furthermore, at baseline and after 6 months of therapy, the functional status of the patients was evaluated using the Health Assessment Questionnaire (HAQ) [18].

### 2.3. Data Analysis

Data were analyzed using SPSS statistics software version SPSS 26.0 (IBM, Armonk, New York, NY, USA). Descriptive statistics were used to characterize the samples. Frequencies and percentages were used to describe categorical variables. Incomplete responses were included in the analysis. The significance of baseline differences was determined by chi-squared test, Fisher’s exact test or unpaired *t*-test as appropriate. A two-sided *p* value < 0.05 was statistically significant. 

Spider plot graphs were produced using the Ggplot2 package (Wickham, 2009) of R studio (R Core Team, Boston, MA, USA, 2014). Graphic analysis of DAS28, NRS and satisfaction questionnaires were produced with Graphpad Prism version 6.04 for Windows (La Jolla, CA, USA).

## 3. Results

A total of 20 patients meeting the inclusion criteria of the study were enrolled in the analysis. In three patients, the treatment was suspended before the completion of the study because of gastrointestinal intolerance; therefore, 17 patients with RA (mean age at inclusion, 59 ± 3 years old; 88% females) completed the study. Table 1 summarizes the demographic and clinical characteristics of the patients who completed follow-up, and Appendix A describes the baseline education level, affinity with technology and personal information.

No differences were detected between the TuTOR mobile app and the PD groups when evaluating demographic, clinical and laboratory parameters. Interestingly, patients felt generally more confident when using a smartphone in comparison to tablet technology or a personal computer (confidence of use on a scale from 1 to 100: 93.3 ± 11.5; 75 ± 23.8; 70 ± 26.5, respectively; *p* < 0.005).

### 3.1. Adherence to Therapy

In order to investigate the compliance and subjective adherence of the enrolled patients, they were asked to fill out specifically designed questionnaires at baseline and at the end of the study after 6 months of therapy.

Figure 2A is a graphical representation of the answers given by the enrolled patients.

It is possible to recognize a global satisfaction and reassurance of the therapy, together with a good subjective compliance to therapy and with an understanding of the severity of the disease. When comparing baseline with 6-month questionnaires, patients reported an amelioration in both mental and physical health and an improved satisfaction with the treatment.

In one of the sections of the adherence questionnaires, we focused on the main reasons for discontinuation of therapy (both occasional and prolonged) and whether and how it changed during the six months of follow-up. Figure 2B shows a graphical representation of the answers of the enrolled patients. As illustrated by the figure, the main reasons for discontinuation were the presence of side effects, subcutaneous administrations and excessive number of tablets to intake. Analyzing baseline questionnaires and comparing the results to those of the questionnaires filled out at the end of the study, the above-mentioned reasons for therapy discontinuation were even more impactful.

### 3.2. DAS28 and NRS Scales

When analyzing progressive DAS28 of the enrolled patients, a statistically significant decrease in DAS28 was observed since the first month of therapy with tofacitinib (mean DAS28 at baseline, 3.9 ± 1 vs. DAS28 at 1 month, 3.1 ± 1; *p* = 0.0016). During the follow-up months, a decreasing trend of DAS28 in all patients was observed (DAS28 at 3rd month, 2.7 ± 1 and at 6th month, 3 ± 1.5) but without a statistically significant reduction when compared to subsequent months. 

When focusing on patients who started with the TuTOR app and comparing them with the PD group, no difference was found in DAS28 at baseline (mean DAS28 of PD group, 4 ± 1.1 vs. TuTOR app DAS28, 4.1 ± 1.1). With respect to the DAS28 during the follow-up months, a similar reduction was observed in the PD group and the TuTOR app group, as assessed with ΔDAS28 (baseline–3 months), (4–6 months) and (baseline–6 months). A statistically significant reduction in DAS28 was observed during the first 3 months when compared to the second half of the study (mean ΔDAS28 baseline-3 months, 1.2 ± 1.2 vs. mean ΔDAS28 4–6 months, 0.3 ± 1.1; *p* = 0.02).

Figure 3 shows the decrease in disease activity assessed by DAS28 in the PD and TuTOR app groups.

In regard to NRS of perceived activity of disease, a statistically significant decrease was observed after the first month of therapy (mean NRS activity at baseline, 5.8 ± 2.1 vs. NRS activity at 1 month, 3.7 ± 2.5; *p* = 0.02). During the follow-up months, a decreasing trend of NRS activity in all patients was observed (NRS at 3rd month, 2.5 ± 1.9 vs. 6th month, 2.6 ± 2) but without reaching a statistically significant reduction. The NRS of perceived activity at baseline was comparable between the TuTOR app and the PD groups, and no difference was found in the NRS activity during the follow-up months. Similarly to the DAS28, during the first period, the decrease in NRS activity was significantly higher (mean ΔNRS activity at 1–3 months, 3.3 ± 6.2 vs. ΔNRS activity at 4–6 months, 1.1 ± 2.3; *p* = 0.03).

When considering NRS subjective pain of the enrolled patients, a decrease was observed after the first month of therapy; however, it did not reach statistical significance (mean NRS pain at baseline, 4.3 ± 2.1 vs. NRS pain at 1 month, 2.5 ± 2 vs. NRS pain at 3rd month, 2.6 ± 1.9 vs. NRS pain at 6th month, 2.5 ± 1.8). Focusing on patients who started with the TuTOR app and comparing them with the PD group, there was a difference between the two group (mean NRS pain in paper diary group, 5.2 ± 1.9 vs. TuTOR app NRS pain, 3 ± 1.6; *p* = 0.024), but the ΔNRS pain at 1–3 months, 1–6 and 4–6 months did not differ between the two groups. As previously observed for NRS pain, the decrease was higher during the first months of follow-up, although without reaching statistical significance (mean ΔNRS pain at 1–3 months, 1.7 ± 2.5 vs. mean ΔNRS pain at 4–6 months, 0.7 ± 2.1).

When analyzing NRS subjective fatigue of the enrolled patients, a statistically significant decrease was observed after the first month of therapy (mean NRS fatigue at baseline, 6.1 ± 2.3 vs. NRS fatigue at 1 month, 4.3 ± 2.6; *p* = 0.01). During the follow-up months, a decreasing trend of NRS fatigue was observed in all patients (NRS fatigue at 3rd month, 3.5 ± 2.5 vs. NRS fatigue at 6th month, 3.4 ± 2.7) but without a statistically significant reduction when compared to subsequent months. The NRS fatigue at baseline was similar between the TuTOR app and PD groups. Furthermore, no differences were observed regarding NRS fatigue in the follow-up period. Similarly to the other scales, the decrease in NRS fatigue was higher in the first part of the study considering all the enrolled patients; however, it did not reach a statistical difference (mean ΔNRS fatigue at 1–3 months, 2.5 ± 2.6 vs. mean ΔNRS fatigue at 4–6 months, 1.5 ± 1.9). Figure 4 shows the decrease in NRS perceived activity, pain and fatigue in the PD and TuTOR app groups.

In terms of which numerical rating scale among activity, fatigue and pain was the most influenced in the follow-up period (assessed by ΔNRS at 1–3 months and 4–6 months), the highest reduction was observed in the ΔNRS at 1–3 months of subjective activity when compared to the other scales (*p* = 0.008).

Additionally, when considering the potential correlations among the DAS28 and NRS scores, a noteworthy correlation was observed between ΔNRS activity and pain (1–3 months, Pearson 0.637; *p* = 0.006; and 4–6 months, Pearson 0.586; *p* = 0.014). 

### 3.3. HAQ

HAQ questionnaires were filled out by the enrolled patients at baseline and after 6 months of follow-up and therapy with tofacitinib. A statistically significant decrease in HAQ was observed during the follow-up months (HAQ at baseline, 1.38 ± 1.11 vs. HAQ at six months, 0.83 ± 0.9; *p* = 0.01). However, despite the significant decrease, the HAQ at 6 months was still higher when compared to the general population [19].

Figure 5 shows the trend of the HAQ at baseline and after 6 months of therapy.

### 3.4. Satisfaction Questionnaire

At the end of study, the patients that completed the follow-up filled out satisfaction questionnaires to gather their subjective opinion on the TuTOR app and the PD (Figure 6).

From what we gathered from the questionnaires, patients seemed to prefer, as an added tool to improve their adherence to treatment, the TuTOR app when compared to the PD.

Whereas the TuTOR app and the PD seemed to be equally handy for the patients, the majority (82%) agreed that the TuTOR app helped them to remember to take the pills (vs. 53% for PD). Furthermore, 82% of patients used the TuTOR app regularly (vs. 53% for PD) and 76% of patients would use it in the future (vs. 53% PD). 

## 4. Discussion

Poor adherence to medication regimens is very common, especially in chronic diseases, and it is responsible for treatment failure, worsening of disease and consequently increased health care cost. Estimates indicate that at least 50% of adults with prescribed therapies for chronic conditions encounter difficulties adhering to their regimen after only six months [10,20].

Adherence is influenced by several factors, such as individual clinical aspects, psychological constructs, therapy-related factors, patient-physician relationship, and external environmental and social factors [21]. Enhancing adherence to treatment could therefore improve the efficacy of medical recommendations and reduce health and financial costs associated with a disease, such as RA, which can lead to premature death and long-term disability [22].

Digital therapeutics are increasing in popularity and usage. They possess the unique ability to grant faster access to health care services and to directly link health care professionals with patients, helping them to achieve better adherence and leading to improved disease management [22].

In recent decades, the therapeutic armamentarium available to rheumatologists has grown considerably, making it possible to control symptoms, to stop radiological progression and to achieve complete remission or at least stable, low disease activity, therefore dramatically changing the natural course of the disease. In this view, improving treatment adherence in RA patients represents a crucial point. Therefore, we developed the TuTOR Mobile app and performed a single-center, prospective controlled study to validate the app on 20 selected patients who jointly began treatment with tofacitinib. The efficacy of the JAK inhibitor was evaluated in previous works [23] and was outside the scope of this article. 

Self-reported adherence was assessed every three months using a specifically designed questionnaire, with each question score going from 1 (never) to 4 (always). The average rate of therapy discontinuation/de-escalation after three months of tofacitinib was 1 ± 0 (never) in both groups (the first group starting with the interventional phase and the second group with the comparative phase). After 6 months of treatment, the average remained 1 ± 0 in the first group and reached 1.43 ± 0.67 in the second group. 

It is well know that self-reported methods of assessing medical compliance often result in an overestimation of adherence [22]. However, with these limitations in mind, the treatment adherence in the first trimester reached 100%, with only a mild decrease during the subsequent three months, which confirms the tendency to discontinue therapy over the months.

As we expected, a high adherence to treatment of our patients was associated with a significant control of disease activity. The subjective perception of disease activity, fatigue and pain was assessed using NRS, resulting in a marked amelioration, especially with respect to the first three months of treatment (see Figure 2B). The same considerations apply to clinically assessed disease activity, which was evaluated using the DAS28 score at every appointment and exhibited a marked decrease during the first 3 months from 3.9 ± 1 at baseline to 3.1 ± 1 at 1 month and 2.7 ± 1 at the 3rd month, (*p* = 0.0016) (Figure 2A). 

We did not observe any difference between the group starting with the interventional phase and the group starting with the comparative phase, suggesting that paper or digital format could achieve the same adherence targets and the same amelioration of disease activity.

When comparing which NRS among activity, fatigue and pain was the most decreased in the follow-up period, the highest reduction was observed in the ΔNRS of subjective activity at 1–3 months (*p* = 0.008). Additionally, when analyzing the potential correlations among the DAS28 and NRS scores, a significant correlation was observed between ΔNRS activity and pain (*p* = 0.014). This might suggest a potential influence of how pain is perceived by the patients as the most impactful component of their condition. 

At the end of both phases of the study, a satisfaction questionnaire was administered to assess the patients’ opinion in relation to the use of the PD or the TuTOR app. What resulted was a general satisfaction with both the supports, with a preference for the TuTOR app for its ability to aid in remembering to take the pills and its ease of use. 

However, our study suffers from some limitations. Importantly, the number of recruited patients was relatively low, and our results should be validated in a larger cohort. Furthermore, questionnaires were filled out directly by our patients, and although more representative of their own subjectivity, the observed results might be difficult to reproduce. It is also important to point out that our study was carried out during the COVID-19 pandemic, which represents a unique scenario and which, in itself, has given great impetus to the use of digital technologies, such as telemedicine, to ensure the continuity of health care assistance, particularly in patients suffering from a chronic disease, such as RA. 

Authors should discuss the results and how they can be interpreted from the perspective of previous studies and of the working hypotheses. The findings and their implications should be discussed in the broadest context possible. Future research directions may also be highlighted.

## 5. Conclusions

Adherence to therapy represents a complex global health and financial problem. We observed that both digital and paper devices can help maximize adherence to therapy, leading to an improvement in control of disease activity. In our study, the TuTOR app was preferred by patients for its simplicity and immediacy. Our study confirmed the importance of filling the need for support expressed by patients, regardless of the tool used.

## Figures and Tables

**Figure 1 ijerph-19-05379-f001:**
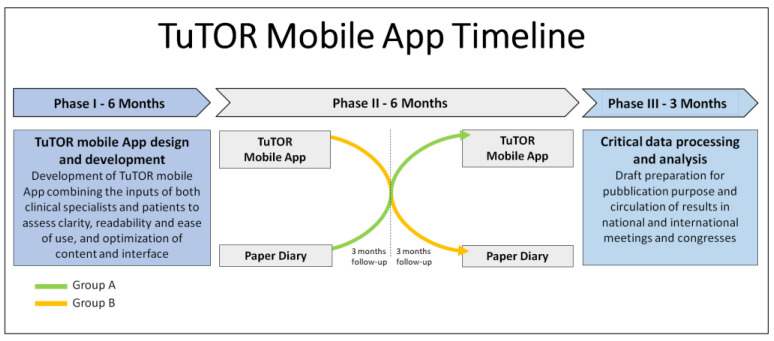
Timeline of the research protocol of the study.

**Figure 2 ijerph-19-05379-f002:**
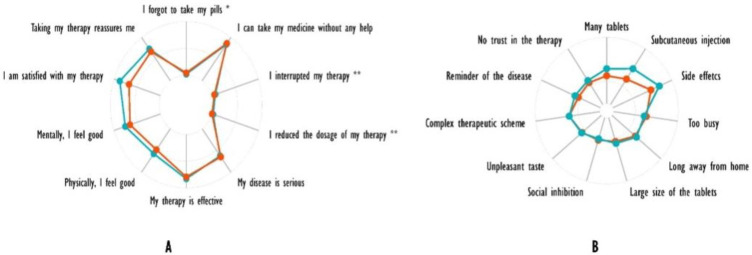
(**A**) Graphical representation of the patients’ reported treatment adherence. (**B**) Graphical representation of the main reason that led patients to discontinue the therapy. * in the last week; ** in the last 4 weeks. **Red**—Baseline; **Blue**—After 6 months.

**Figure 3 ijerph-19-05379-f003:**
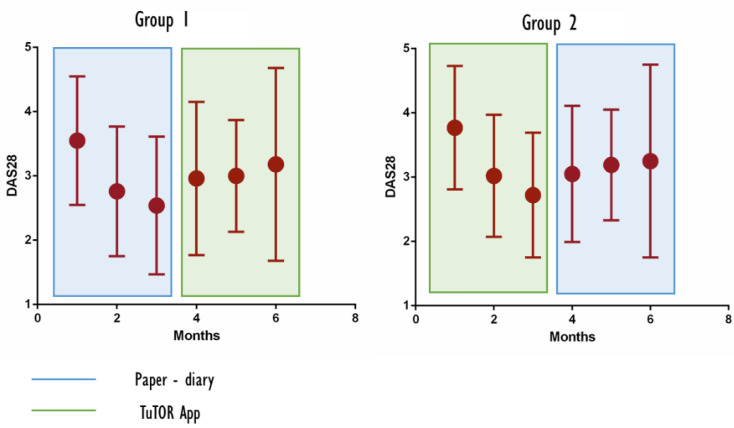
Decrease in disease activity assessed by DAS28 in the paper diary and TuTOR app groups.

**Figure 4 ijerph-19-05379-f004:**
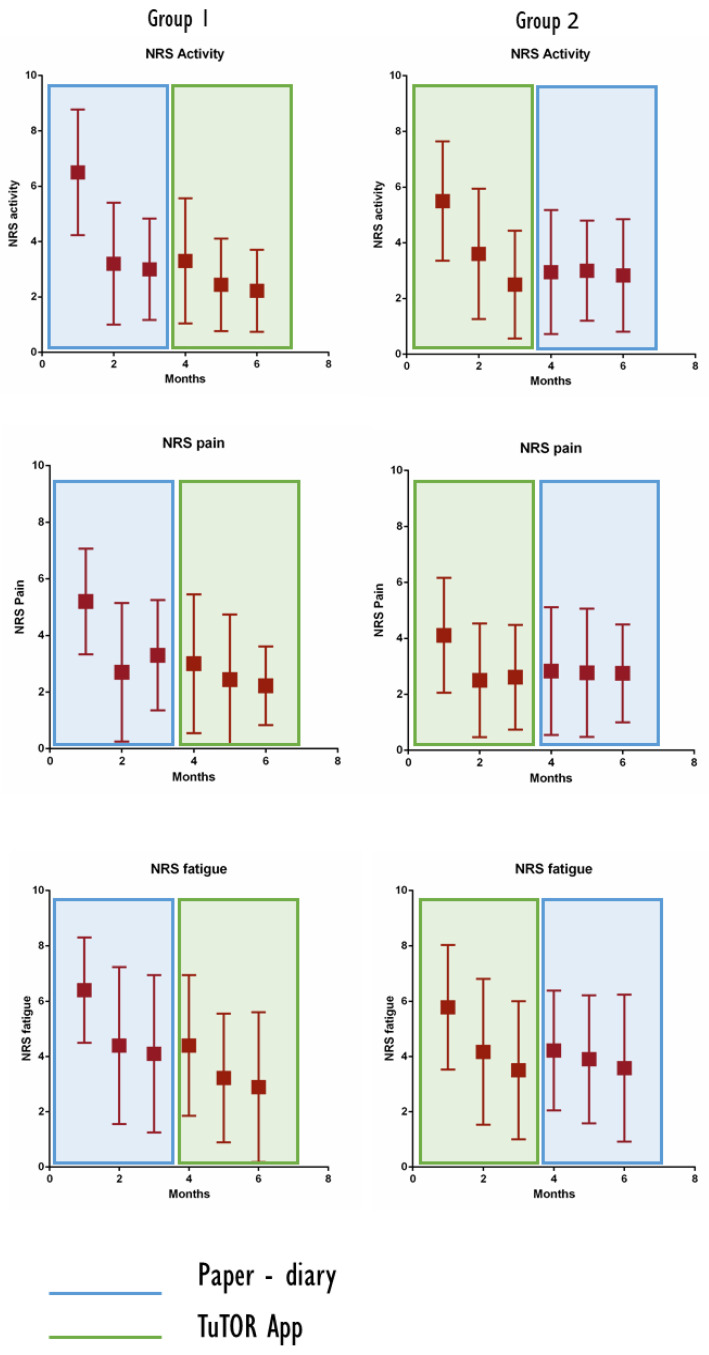
Decrease in NRS perceived activity, pain and fatigue in the PD and TuTOR app groups.

**Figure 5 ijerph-19-05379-f005:**
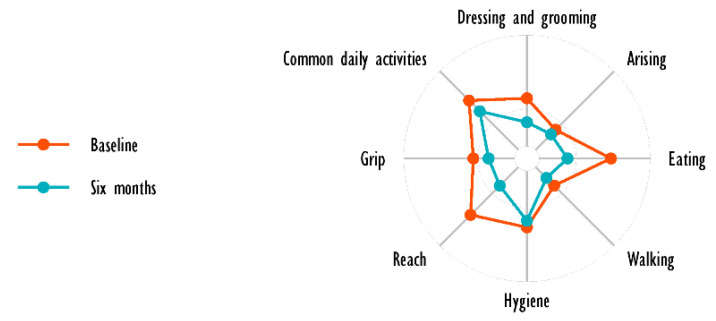
Trend of the HAQ at baseline and after 6 months of therapy.

**Figure 6 ijerph-19-05379-f006:**
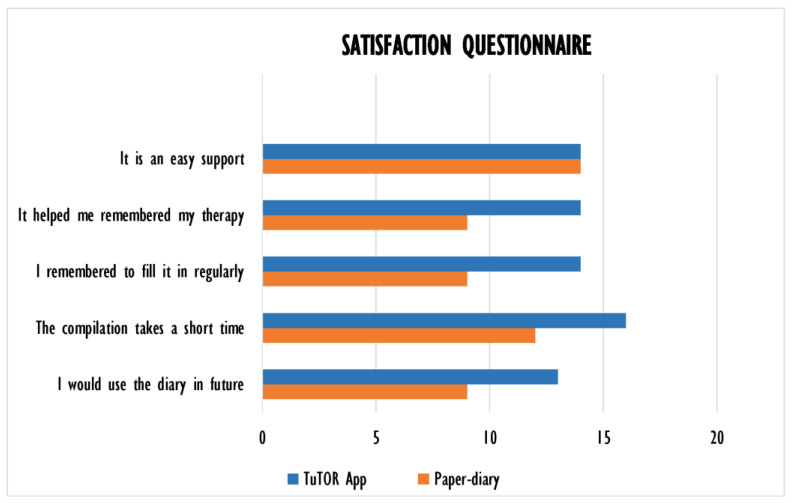
Graphical representation of the results of the satisfaction questionnaires filled out at the end of the study by the enrolled patients.

**Table 1 ijerph-19-05379-t001:** Baseline demographic and clinical characteristics of the patients enrolled in the study who completed follow-up. RA—rheumatoid Arthritis.

	Patients with RA (N = 17)
**DEMOGRAPHICS**	
Age (mean ± S.D.)	59.4 ± 13.5
Sex (n; %)	M (2; 11.8). F (15; 88.2)
Ethnicity (Caucasian; n; %)	16; 94
Ethnicity (Hispanic; n; %)	1; 6
**CLINICAL CHARACTERISTICS**	
Age at diagnosis (mean ± S.D.)	44.7 ± 14.47
Follow-up length (years; mean ± S.D.)	6.33 ± 5.17
Positive rheumatoid factor (n; %)	17; 100
Positive anti-cyclic citrullinated peptides (n;%)	15; 88.2
Structural articular damage at radiography (n; %)	11; 7

## Data Availability

Not applicable.

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
