# Peer review of "Tailoring Tofacitinib Oral Therapy in Rheumatoid Arthritis: The TuTOR App"

_ijerph, 2022, doi:10.3390/ijerph19095379_

Round 1

Reviewer 1 Report

I would like to get more explanation on

"All patients at baseline, through the
medium of certified questionnaires based on digital health literacy, had been evaluated to
assess the affinity for technology and its influence on the outcomes of the study and sub-
sequently received a personal learning-by-doing tutorial session.

The overall use (expertise and practice) of tofacitinib befor the start of the study in this particular setting (hospital or out patient clinic).

Small sample size and not justified. Some explanations needed.

Author Response

Reviewer 1

I would like to get more explanation on:

Reviewer 1 Point 1

All patients at baseline, through the medium of certified questionnaires based on digital health literacy, had been evaluated to assess the affinity for technology and its influence on the outcomes of the study and subsequently received a personal learning-by-doing tutorial session.

Reviewer 1 Reply 1

Thank you for the comment. Data concerning the patients’ affinity for technology, obtained with the certified questionnaires, are showed in Table S1 in the supplementary material.

Since this information was lacking in the manuscript, it has been added in the section Materials and Methods - Study Design, to read “All patients at baseline had been evaluated to assess the affinity for technology and its influence on the outcomes of the study through the medium of certified questionnaires based on digital health literacy. Subsequently they received a personal learning-by-doing tutorial session in order to be familiarized with the functions of the application (i.e., confirming medication intake and recording follow-up visits and clinical appointments). Table S1 shows the baseline educational level, affinity with technology and personal information of the patients enrolled in the study.

Reviewer 1 Point 2

The overall use (expertise and practice) of Tofacitinib before the start of the study in this particular setting (hospital or out-patient clinic).

Reviewer 1 Reply 2

Thank you for the comment. Tofacitinib is a biological agent that is approved for the treatment of RA and that is normally use in the rheumatologic clinical practice. Tofacitinib was chosen in this study because it has the peculiarity, when compared to most biological agents, to be administered orally and, importantly, daily. The patients must therefore take the drug every day, and compliance plays a huge role in determining response to treatment. 

Reviewer 1 Point 3

Small sample size and not justified. Some explanations needed.

Reviewer 1 Reply 3

Thank you for the comment. We do agree with the reviewer that sample size was limited when considering our study, as stated in the limitation section: “Importantly, the number of recruited patients was relatively low, and our results should be validated in a larger cohort”. The main objective of this study was to validate the TuTOR App and to start testing its applicability to a small group of patients in order to further implement its use in a hematological setting, possibly to a larger number of patients suffering from different conditions and taking different treatments.

Reviewer 2 Report

This is an interesting work about developing a mobile app for Tailoring Tofacitinib Oral therapy in Rheumatoid arthritis which may be accepted after major revision.

  1. How many patients were taken in each inclusion and exclusion group, need to be mentioned in methods?
  2. Why females ratio is high compared to male patients, any specific reason
  3. The quality of Figure 2 is poor, needs to be improved with better pixels.
  4.  Line 379-382 appears to be statements from the reviewer, please check and corrected them accordingly.
  5. There are many grammatical errors that need to be corrected

Author Response

Reviewer 2

This is an interesting work about developing a mobile app for Tailoring Tofacitinib Oral therapy in Rheumatoid arthritis which may be accepted after major revision.

Reviewer 2 Point 1

How many patients were taken in each inclusion and exclusion group, need to be mentioned in methods?

Reviewer 2 Reply 1

Thank you for the comment. As stated in the manuscript, a total of 20 patients met the inclusion criteria of the study and started using the TuTOR App and the treatment with Tofacinib, however in three cases the treatment was interrupted due to GI intolerance and were excluded from the study. To read: “In three patients the treatment was suspended before the completion of the study because of gastrointestinal intolerance, therefore 17 patients with RA (mean age at inclusion 59 ±13-years-old; 88% females) completed the study.”

Reviewer 2 Point 2

Why females ratio is high compared to male patients, any specific reason

Reviewer 2 Reply 2

Thank you for the comment. It’s well known that the prevalence of RA is higher in females than males. In particular, the incidence in female has been reported to be 4-5 times higher below the age of 50, decreasing to 2-3 time higher when considering age above 65 years (Myasoedova E et al, Is the incidence of rheumatoid arthritis rising?: results from Olmsted County, Minnesota, 1955-2007. Arthritis Rheum. 2010 Jun). Taken the above in consideration, it results that the female percentage of 88% of our study is perfectly in line with the previous results reported in literature.

Reviewer 2 Point 3

The quality of Figure 2 is poor, needs to be improved with better pixels.

Reviewer 2 Reply 3

Apologies for the inconvenience. Figure 2 has been uploaded in the manuscript in a better resolution format to ensure its readability.

Reviewer 2 Point 4

Line 379-382 appears to be statements from the reviewer, please check and corrected them accordingly.

Reviewer 2 Reply 4

Apologies for the inconvenience. It was a typo of the Word template file used for the submission system. We have checked and deleted the paragraph.

Reviewer 2 Point 5

There are many grammatical errors that need to be corrected.

Reviewer 2 Reply 5

Thank you for the suggestion. The manuscript has been carefully revised by an English native speaker in order to avoid any errors. The manuscript has been double-checked for typos.

Reviewer 3 Report

The authors continue their research on the designed application - the TuTOR (Tailoring Tofacitinib Oral therapy in Rheumatoid arthritis) Mobile App. In the presented article they assessed the effect of application on adherence to tofacitinib therapy in patients with rheumatoid arthritis. The idea and study is quite interesting and clinically important.

However there are some issues that need to be addressed: 

  • Please use the form of citation appropriate for the journal
  • Although the results do not indicate differences in adherence to therapy in people using the application, the main question arises - is the presented study group sufficient to obtain reliable results. What is the power of the statistical test and what is the estimated minimum group on which to show differences.
  • The next biggest problem is the term adherence to treatment. The self-reported method is not objective. In clinical trials, the calculation of the number of tablets dispensed and returned is used, possibly the assessment of drug concentration in the blood. Since the basis for the creation of the application was the problem with patient compliance, this parameter cannot be assessed solely on the basis of information from patients.
  • Taking into account most of the limitations of the study, in particular the small number of patients, the lack of objective assessments of adherence to therapy, the obtained results are burdened with high uncertainty, which does not allow to unequivocally confirm or deny the effectiveness of the application. And this was to be the purpose of the presented work. The idea of the application was presented in the previous publication.

Author Response

Reviewer 3

The authors continue their research on the designed application - the TuTOR (Tailoring Tofacitinib Oral therapy in Rheumatoid arthritis) Mobile App. In the presented article they assessed the effect of application on adherence to tofacitinib therapy in patients with rheumatoid arthritis. The idea and study is quite interesting and clinically important.However, there are some issues that need to be addressed: 

Reviewer 3 Point 1

Please use the form of citation appropriate for the journal

Reviewer 3 Reply 1

Thank you for the comment. The citation style has been modified accordingly to the specific directions of the journal.

Reviewer 3 Point 2

Although the results do not indicate differences in adherence to therapy in people using the application, the main question arises - is the presented study group sufficient to obtain reliable results. What is the power of the statistical test and what is the estimated minimum group on which to show differences.

Reviewer 3 Reply 2

Thank you for the comment.

We do agree with the reviewer that sample size was limited when considering our study, as stated in the limitation section: “Importantly, the number of recruited patients was relatively low, and our results should be validated in a larger cohort”. Indeed, while the TuTOR App might in fact be more useful than the paper diary, the results of the present study might not be able to show it due to the low number of the included patients. However, the main objective of this study was to validate the TuTOR App and to start testing its applicability to a small group of patients in order to further implement its use in a rheumatological setting, possibly to a larger number of patients suffering from different conditions and taking different treatments.

Reviewer 3 Point 3

The next biggest problem is the term adherence to treatment. The self-reported method is not objective. In clinical trials, the calculation of the number of tablets dispensed and returned is used, possibly the assessment of drug concentration in the blood. Since the basis for the creation of the application was the problem with patient compliance, this parameter cannot be assessed solely on the basis of information from patients.

Reviewer 3 Reply 3

Thank you for the comment. The treatment’s adherence was assessed both with the self-reported method and with the pills count. This aspect is now adequately emphasized in the manuscript and we thank the reviewer for having pointing this out. The information has been added in the section Material and Methods – Study Design, to read: “Tablets were supplied in blisters of 56 film-coated tablets. At each visit, patients were required to bring their medication blisters: the number of remaining tablets was recorded by the clinician and compared with the self-reported adherence.”

Reviewer 3 Point 4

Taking into account most of the limitations of the study, in particular the small number of patients, the lack of objective assessments of adherence to therapy, the obtained results are burdened with high uncertainty, which does not allow to unequivocally confirm or deny the effectiveness of the application. And this was to be the purpose of the presented work. The idea of the application was presented in the previous publication.

Reviewer 3 Reply 4

Thank you for the comment. We appreciate the precious suggestions. In this revised version, the main comments of the reviewers have been addressed and limitations of the study clearly acknowledged.

All in all, we feel we met the main objective of our study, that was in fact to validate the TuTOR App and to start pivotally testing its applicability to a small group of patients in order to further implement its use in a rheumatological setting. Once tested, its use should be explored in a larger number of patients suffering from  different conditions and taking different treatments.

For the sake of clarity, the objective of the study has been rephrased, to read: “Our primary objective was to validate the use of the TuTOR App in a controlled population of RA patients and, secondly, whether the use of a mobile application could ameliorate therapy’s adherence”.

Round 2

Reviewer 2 Report

The revised manuscript have been improved and may be considered for the publication in the present form. 

Reviewer 3 Report

The authors took into account most of the reviewers' comments, significantly bringing up the manuscript. However, the question remains whether, with such a small group, the results are reliable - which calls into question the sense of the publication of the entire work in good journal as original study.